# Population genomics of louping ill virus provide new insights into the evolution of tick-borne flaviviruses

Jordan J. Clark[1,2]*, Janice Gilray[2], Richard J. Orton[1], Margaret Baird[1], Gavin Wilkie[1], Ana da Silva Filipe[1], Nicholas Johnson[3,4], Colin J. McInnes[2], Alain Kohl[1], Roman Biek[5]*

**1** MRC-University of Glasgow Centre for Virus Research, Glasgow, United Kingdom, **2** Moredun Research Institute, Edinburgh, United Kingdom, **3** Animal and Plant Health Agency, Addlestone, Surrey, United Kingdom, **4** Faculty of Health and Medical Science, University of Surrey, Guildford, Surrey, United Kingdom, **5** Institute of Biodiversity, Animal Health and Comparative Medicine - University of Glasgow, Glasgow, United Kingdom

* jordan.clark@liverpool.ac.uk (JC); Roman.Biek@glasgow.ac.uk (RB)

**Data Availability Statement:** The genome sequences which were produced as part of the manuscript are available from GenBank (https://www.ncbi.nlm.nih.gov/genbank/). Accession

## Abstract

The emergence and spread of tick-borne arboviruses pose an increased challenge to human and animal health. In Europe this is demonstrated by the increasingly wide distribution of tick-borne encephalitis virus (TBEV, *Flavivirus*, *Flaviviridae*), which has recently been found in the United Kingdom (UK). However, much less is known about other tick-borne flaviviruses (TBFV), such as the closely related louping ill virus (LIV), an animal pathogen which is endemic to the UK and Ireland, but which has been detected in other parts of Europe including Scandinavia and Russia. The emergence and potential spatial overlap of these viruses necessitates improved understanding of LIV genomic diversity, geographic spread and evolutionary history. We sequenced a virus archive composed of 22 LIV isolates which had been sampled throughout the UK over a period of over 80 years. Combining this dataset with published virus sequences, we detected no sign of recombination and found low diversity and limited evidence for positive selection in the LIV genome. Phylogenetic analysis provided evidence of geographic clustering as well as long-distance movement, including movement events that appear recent. However, despite genomic data and an 80-year time span, we found that the data contained insufficient temporal signal to reliably estimate a molecular clock rate for LIV. Additional analyses revealed that this also applied to TBEV, albeit to a lesser extent, pointing to a general problem with phylogenetic dating for TBFV. The 22 LIV genomes generated during this study provide a more reliable LIV phylogeny, improving our knowledge of the evolution of tick-borne flaviviruses. Our inability to estimate a molecular clock rate for both LIV and TBEV suggests that temporal calibration of tick-borne flavivirus evolution should be interpreted with caution and highlight a unique aspect of these viruses which may be explained by their reliance on tick vectors.

numbers: MK007551, MK007532, MK007550, MK007545, MK007544, MK007549, MK007538, MK007548, MK007547, MK007546, MK007543 MK007542, MK007541, MK007540, MK007539, MK007537, MK007536, MK007535, MK007534, MK007533, MN844186, MH537791. The raw FASTQ sequence reads have been submitted to the Europe Nucleotide Archive (ENA) under accession number PRJEB38554.

**Funding:** JC is funded by UK Biotechnology and Biological Sciences Research Council WestBio Doctoral Training Partnership 2012 (BB/J013854/1) https://bbsrc.ukri.org/. RJO and AK are supported by the UK Medical Research Council with grants (MC_UU_12014/12) and (MC_UU_12014/8), respectively https://mrc.ukri.org/. JG and CJM are funded through the Strategic Research Programme of the Rural and Environmental Science and Analytical Services Division of the Scottish Government. NJ is supported by the Department for Environment, Food and Rural Affairs, and the Scottish and Welsh Governments (SV3045) https://www.gov.uk/government/organisations/department-for-environment-food-rural-affairs. The funders had no role in study design, data collection and analysis, decision to publish, or preparation of the manuscript.

**Competing interests:** The authors have declared that no competing interests exist.

## Author summary

Tick-borne pathogens represent a major emerging threat to public health and in recent years have been expanding into new areas. LIV is a neglected virus endemic to the UK and Ireland (though it has been detected in Scandinavia and Russia) which is closely related to the major human pathogen TBEV, but predominantly causes disease in sheep and grouse. The recent detection of TBEV in the UK, which has also emerged elsewhere in Europe, requires more detailed understanding of the spread and sequence diversity of LIV. This could be important for diagnosis and vaccination, but also to improve our understanding of the evolution and emergence of these tick-borne viruses. Here we describe the sequencing of 22 LIV isolates which have been sampled from several host species across the past century. We have utilised this dataset to investigate the evolutionary pressures that LIV is subjected to and have explored the evolution of LIV using phylogenetic analysis. Crucially we were unable to estimate a reliable molecular clock rate for LIV and found that this problem also extends to a larger phylogeny of TBEV sequences. This work highlights a previously unknown caveat of tick-borne flavivirus evolutionary analysis which may be important for understanding the evolution of these important pathogens.

## Introduction

The tick-borne encephalitis sub-complex of the genus *Flavivirus* (family *Flaviviridae*) is composed of several closely related arboviruses, all of which possess a single-stranded, positive sense, RNA genome of ~12kb [1]. Members of this subcomplex are found across the northern hemisphere [2,3] and include several zoonotic viruses, such as tick-borne encephalitis virus (TBEV), of which there are three predominant subtypes: European (TBEV-Eu), Siberian (TBEV-Sib) and Far-Eastern (TBEV-FE). TBEV frequently infects humans as incidental hosts, resulting in febrile illness and often fatal encephalitis [4]. The case fatality rate (CFR) of TBEV varies according to subtype, with TBEV-Eu and TBEV-Sib exhibiting a CFR of 1–2%, while the highly pathogenic TBEV-FE subtype exhibits a CFR of 20–40% [5]. In contrast, louping ill virus (LIV), which is closely related to TBEV, is predominantly associated with disease in ruminants and birds, whereas human cases are rare [6,7].

LIV is endemic to the British Isles [8] and spread by the hard tick *Ixodes ricinus*. First isolated in the early 1930s, LIV was identified as the causative agent of louping ill disease, which can cause mortalities in sheep (*Ovis aries*) and, more significantly, red grouse (*Lagopus lagopus scotica*) [9,10]. LIV is of significant economic concern to sheep farmers and the game estates where grouse are maintained for commercial shooting purposes. The threat posed by LIV is further aggravated by the movement of tick populations into new regions and higher altitudes [11–13], and a reduction in tick inter-stadial development time due to climate change [14]. While LIV is predominantly found within the UK and Republic of Ireland, the virus has also been reported in Southern Norway [15], the Danish island of Bornholm [16], and Far-Eastern Russia [17]. However, the biological mechanisms responsible for this disjointed geographic distribution are not well understood. Whereas the genomic diversity of TBEV is well characterised, evolutionary studies of LIV have been limited to the analysis of single genes [18], which can lead to inconsistent phylogenetic inference [19,20]. At present, only four complete LIV genomes are available, which has prevented a meaningful analysis of LIV genomic diversity and evolution and their comparison with other flaviviruses.

Whilst recombination has previously been shown to contribute to dengue virus [21,22] and Japanese encephalitis virus evolution [23], recombination in the tick-borne flaviviruses (TBFV) is controversial. Two studies have reported a recombination event between LIV and TBEV-Eu [20,24]; however, subsequent work has raised doubts about these findings [25]. Although the geographic ranges of the two viruses are largely distinct [26], creating limited opportunities for interaction, recent detection of TBEV in Britain [27–29], where LIV is endemic, has added new urgency to investigate the possibility of recombination between the two viruses.

Phylogenies are essential for determining the epidemiological and evolutionary history of viruses, and, by calibrating trees with a molecular clock, to place their history into a temporal context [30]. Previous phylogenetic studies have reported clock rates and time calibrated trees for both TBEV and LIV; however, only in the case of TBEV were estimates based on full genomes [20,31,32]. In contrast, rates for LIV were derived from glycoprotein sequences, resulting in considerable uncertainty with respect to phylogenetic relationships and divergence times [18]. Moreover, previous studies in both viruses implicitly assumed that time-stamped sequence data sets contained sufficient evolutionary signal to estimate clocks reliably. Recent work on other RNA viruses has shown that this assumption is not always met [33], warranting a more careful approach to phylogenetic dating in TBFV.

Here, we address these knowledge gaps based on a novel dataset of 22 LIV genomes derived from UK isolates sampled over the past century. Combining these data with published genome sequences for LIV (Fig 1) and related viruses we sought to i) conduct the first systematic test for recombination and positive selection in LIV, ii) clarify the phylogeography of LIV and its phylogenetic relationship to other TBEV, and iii) re-assess the evolutionary rate and divergence time estimates of LIV and TBEV.

## Methods

### Virus isolates

The isolation of the virus isolates sequenced in this study, with the exception of ENG_PEN6_2009, and WA_AB2_2010, is described elsewhere [18]. ENG_Dog_2015 was isolated from the brain of an infected dog and has been previously described [34]. All viruses with the exception of ENG_PEN6_2009, and WA_AB2_2010 were passaged once in BHK21 cells (which were derived from cell stocks present at the Moredun Research Institute). Isolates ENG_A_1980, ENG_DEV1_1983, ENG_DEV2_1989, IRE_IRE3_1968, and SCO_LOCH2_1993, did not sequence well after passage in BHK21 cells, therefore they were grown in the cell line CPT-Tert [35]. The CPT-Terts are an ovine derived cell line which were kindly provided by Dr. David Griffiths (Moredun Research Institute). All cells were maintained in DMEM supplemented with 10% FCS, 1% HEPES, 1000 units/ml penicillin and 1 mg/ml streptomycin in T25 cell culture flasks. Cells were infected at a MOI of 0.1 and incubated at 37°C with a 5% $CO_2$ atmosphere for 36–48 hours at which point CPE became obvious.

### RNA extraction & sequencing

RNA extraction of cells in 25 $cm^2$ flasks was carried out using TRIzol reagent (Thermo Fisher Scientific). Cell culture media was removed, cells washed twice with PBS, and 1 ml TRIzol reagent added to the cell monolayer. Samples ENG_PEN6_2009 and WA_AB2_2010 were obtained from sheep brain tissue which was taken from suspected LIV cases and homogenised in TRIzol reagent (Thermo Fisher Scientific). RNA extraction was carried out as per manufacturer's instructions. RNA libraries were prepared for sequencing using the TruSeq RNA kit (Illumina), following the manufacturer's protocol. Briefly, sample preparation utilized 400 ng

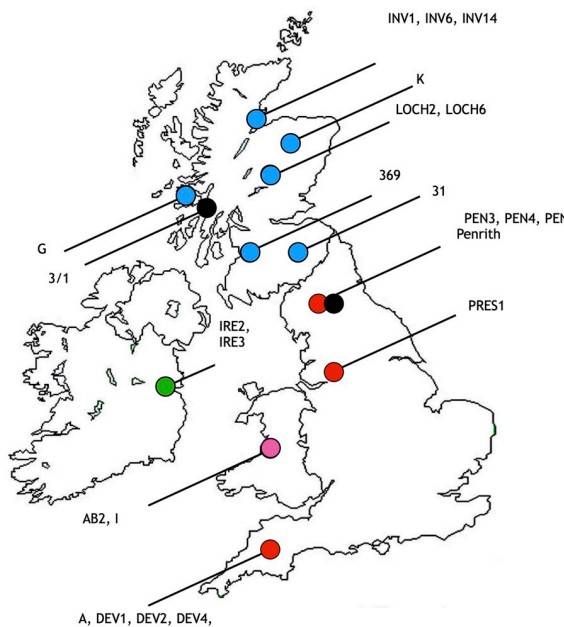

| Isolate | Year of isolation | Host Species | Sampling Location |
|---|---|---|---|
| SCO_31_1931 | 1931 | *Ovis ares* | Selkirk (Scotland) |
| LI3/1 | 1962 | *Ovis ares* | Oban (Scotland) |
| SCO_369/T2_1963 | 1963 | *Ixodes ricinus* | Ayrshire, Scotland |
| SCO_G_1979 | 1979 | *Sus domesticus* | Isle of Mull (Scotland) |
| SCO_K_1980 | 1980 | *Lagopus lagopus scotica* | Grampians (Scotland) |
| SCO_INV1_1983 | 1983 | *Ovis ares* | Inverness (Scotland) |
| SCO_INV6_1986 | 1986 | *Ovis ares* | Inverness (Scotland) |
| SCO_INV14_1992 | 1992 | *Ovis ares* | Inverness (Scotland) |
| SCO_LOCH2_1993 | 1993 | *Lagopus lagopus scotica* | Lochindorb (Scotland) |
| SCO_LOCH6_1993 | 1993 | *Lagopus lagopus scotica* | Lochindorb (Scotland) |
| ENG_A_1980 | 1980 | *Ovis ares* | Devon (England) |
| ENG_PEN3_1983 | 1983 | *Ovis ares* | Penrith (England) |
| ENG_PEN4_1983 | 1983 | *Ovis ares* | Penrith (England) |
| ENG_DEV1_1983 | 1983 | *Ovis ares* | Devon (England) |
| ENG_DEV2_1989 | 1989 | *Ovis ares* | Devon (England) |
| ENG_PRES1_1991 | 1991 | *Ovis ares* | Preston (England) |
| ENG_DEV4_1995 | 1995 | *Ovis ares* | Devon (England) |
| ENG_PEN6_2009 | 2009 | *Ovis ares* | Penrith (England) |
| Penrith | 2009 | *Ovis ares* | Penrith (England) |
| ENG_Dog_2015 | 2015 | *Canis lupus familiaris* | Devon (England) |
| IRE_IRE3_1968 | 1968 | *Ovis ares* | Dublin (Ireland) |
| IRE_IRE2_1971 | 1971 | *Ixodes ricinus* | Dublin (Ireland) |
| WA_I_1980 | 1980 | *Ovis ares* | Aberystwyth (Wales) |
| WA_AB2_2010 | 2010 | *Ovis ares* | Aberystwyth (Wales) |
| Primorye-185-91 | 1991 | *Homo sapiens* | Vladivostok (Russia) |
| LEIV-7435Tur | 1985 | *Hyalomma marginatum* | Turkmenistan |

**Fig 1. Map of the UK showing the sampling locations of the 22 LIV isolates generated in this study.** Isolates which have been sequenced during this study are coloured according to their country of isolation, with Scotland shown as blue, England as red, Ireland as green, and Wales as pink. Isolates whose sequence has been downloaded from GenBank are shown as black circles. Strains Primorye-185-91 and LEIV-7435Tur were isolated in the Russian Far-East and Turkmenistan and are therefore not shown on the map. Further details of the 26 LIV genomes utilised in this study are also presented.

of sample RNA, measured using a Qubit 3.0 fluorometer (Invitrogen). RNA was chemically fragmented and first strand cDNA generated using SuperScript II (Thermo Fisher Scientific) as per manufacturer's instructions. Following the ligation of index sequences, samples were cleaned using AMPure XP beads (Beckman Coulter) and washed twice in 70% (v/v) ethanol.

Following resuspension PCR was employed to enrich for the cDNA fragments which had adapter sequences ligated to their ends as per manufacturer's instructions. The thermal cycler conditions employed were as follows: 98˚C for 30 sec followed by 15 cycles of 98˚C for 10 sec, 60˚C for 30 sec, 72˚C for 30 sec, and a final extension of 72˚C for 5 minutes. Following PCR, libraries were cleaned with AMPure XP beads as described before.

All libraries were pooled together at equimolar concentrations. The molarity of the libraries was calculated based on mass concentration using a Qubit 3.0 fluorometer (Invitrogen) and the size of the fragments using an Agilent 2200 TapeStation (Agilent) as per manufacturer's instructions. Pooled libraries were then sequenced on an Illumina MiSeq (Illumina). This sequencing strategy utilises paired end reads of 250bp. The number of reads obtained for each genome and the genome coverage of each genome is displayed in S1 Table.

## Sequence assembly

Prior to bioinformatic analysis, read quality was assessed using FASTQC (http://www. bioinformatics.babraham.ac.uk/projects/fastqc/). The raw FASTQ sequence reads have been submitted to the Europe Nucleotide Archive (ENA) under accession number PRJEB38554. Adapter sequences were removed, and quality filtered using trim_galore (https://www. bioinformatics.babraham.ac.uk/projects/trim_galore/) utilising a quality threshold of Q25 and a minimum read length of 75. Reads were also filtered for low complexity and duplicates using prinseq [36]. Filtered reads were subsequently mapped onto a LIV complete genome sequence

downloaded from GenBank (LIV 369/T2, accession number: Y07863) using alignment software (BWA-MEM, [37]). The assembled data was parsed using DiversiTools (http://josephhughes.github.io/btctools/) to determine the frequency of nucleotides at each site and to construct a consensus sequence; consensus is defined as the most dominant base at each genome position. To validate the consensus sequences generated via read alignment to the reference genome, all sequences from the first twelve samples we sequenced were also *de novo* assembled using SPAdes [38]. Reference aligned sequences and *de novo* assembled sequences were found to be identical, therefore all subsequent genomes were generated via reference alignment. The accession numbers corresponding to the sequences generated during this study are shown in S2 Table.

### Sequence alignment & analysis

Multiple sequence alignment was carried out using the MUSCLE program within the Geneious software package [39,40]. Eight iterations were utilised within MUSCLE. Distance tables were generated within Geneious to determine the pairwise identity of sequences. The sequences utilised for this analysis are shown in S2 Table.

### Recombination screening

Alignments were utilised for recombination analysis using recombination detection program 4 (RDP4) [41]. Within RDP4 the: RDP, geneconv, Bootscan, MaxChi, Chimaera, Siscan and 3Seq methods were utilised [42–48]. Standard parameters were employed for screening. Recombination events with a p-value of <0.05 and which were identified by more than three methods were considered significant.

### Selection analysis

Selection analysis was carried out using the HYPHY software via the online selection tool Datamonkey (http://www.datamonkey.org/) [49,50]. Tests for positive selection were carried out using the fixed effects likelihood (FEL) analysis, and Mixed Effects Model of Evolution (MEME) analysis [50,51]. Genome-wide selection was performed using an alignment of 26 LIV coding sequences (CDS) however, single gene alignments were also analysed. For this analysis, a p-value of ≤ 0.05 was considered significant. HYPHY was also utilised to estimate a phylogenetic tree using the dataset of 26 LIV genomes and the aBS-REL (adaptive branch-site random effects likelihood) model [49,50,52]. The resulting tree was downloaded from the Datamonkey website and the total branch length compared with the total branch length of the ML tree generated using PHYML using the ape package within the R software package (R Core Team, https://www.r-project.org/, http://ape-package.ird.fr/).

### Phylogenetic analysis

The most suitable substitution model for the LIV dataset investigated using the program JModelTest2 (https://github.com/ddarriba/jmodeltest2 [53]). The most suitable model with the lowest BIC score was found to be the GTR+G+I model. Bayesian phylogenetic trees were generated using the program MrBayes within the Geneious software suite [40,54]. The GTR+G+I model was employed with 4 gamma categories and an MCMC chain length of 1 million with 4 heated chains. 10% of these were discarded as burn-in and a consensus tree generated. Omsk haemorrhagic fever virus (accession number: NC_005062) was included as an outgroup.

Maximum likelihood [ML] trees were utilised for root-to-tip divergence analysis of the LIV dataset and were generated using the program PHYML within the Geneious software suite

[40,55]. Trees were estimated using the GTR substitution model with 1000 parametric boot-strap replicates. Root-to-tip divergence analysis to test for the presence of a molecular clock in the LIV phylogeny was carried out using the program TempEst utilising the best-fitting root option (http://tree.bio.ed.ac.uk/software/tempest/)[56]. The rate of evolution of LIV was estimated using the Bayesian evolutionary analysis sampling trees (BEAST) software package (http://beast.community/, version 1.8.4) [57]. To determine the most appropriate molecular clock model and coalescent model for the LIV dataset, marginal likelihood estimation (MLE) using path-sampling (PS) and stepping-stone (SS) was carried out [57–59]. A chain length of 50,000,000 was utilised with 100 steps of chain length 1,000,000 for the PS or SS MLE. A model with a lognormal relaxed clock and a Bayesian skygrid tree prior was found to have the lowest MLE and was therefore deemed most suitable [60,61].

BEAST analysis was carried out using the GTR+G+I substitution model, the uncorrelated lognormal relaxed clock, the Bayesian skygrid coalescent model and a chain length of 50,000,000. The BEAGLE library was employed when running BEAST to improve the speed of runs [62]. The resulting log files were analysed using the program Tracer (https://github.com/beast-dev/tracer/releases/tag/v1.7.1, version 1.7.0, [63]) to ensure that the effective sample size (ESS) of each parameter in the complete BEAST run was above 200. Final maximum clade credibility (MCC) trees were generated using the program TreeAnnotator (version 1.8.4) with burn-in specified as 10% of states. To determine the validity of the clock estimate, BEAST analysis was repeated using a null model composed of the original sequence data with dates randomised. This analysis was repeated 20 times and the clock rate estimates compared to the estimates produced utilising the heterochronus dataset using the program TRACER. A TBE-V-Eu tree was also generated in BEAST using an alignment of 36 TBEV-Eu sequences sampled across Europe and Russia. A chain length of 50,000,000 and the Bayesian skygrid coalescent model was employed. As with the LIV dataset, the clock-rate estimate was investigated using 20 date randomised datasets. All final phylogenetic trees were visualised using FigTree (version 1.4.3, http://tree.bio.ed.ac.uk/software/figtree/).

## Results

### No evidence for recombination in LIV

New complete genome sequences of 22 LIV isolates were obtained and aligned with 4 LIV genomes available from GenBank (Fig 1), as well as 36 TBEV-Eu genomes, four TBEV-Sib genomes, 21 TBEV-FE genomes and single genomes of Spanish sheep encephalitis virus (SSEV), Spanish goat encephalitis virus (SGEV), Greek goat encephalitis virus (GGEV), and Turkish sheep encephalitis virus (TSEV) (total of 91 TBFV genome sequences; S1 Table). Incorporation of the published LIV 369(T2) sequence (GenBank accession number Y07863, hereafter referred to as LIV 369/T2_Y07863) in this alignment resulted in the detection of a recombination signal located between nucleotides 5924 and 6129 of the LIV genome (Fig 2A). However, our dataset of 22 newly generated LIV genomic sequences included a re-sequencing of the original LIV 392(T2) isolate (hereafter referred to as SCO_369/T2_1963), which we found to differ from that published in GenBank. When the published sequence was replaced with the newly generated sequence, the recombination signal was no longer detected. Aligning the recombinant region containing both the published and re-sequenced LIV 369(T2) isolate, reveals that bases 5924 and 6129 of the LIV genome are shared between the GenBank derived LIV 369/T2_Y07863 sequence and several TBEV-Eu strains, whilst in the re-sequenced SCO_369/T2_1963 genome, this region is highly similar to all other LIV isolates (Fig 2B). This suggests that that the recombination signal detected in the published LIV 369/T2_Y07863

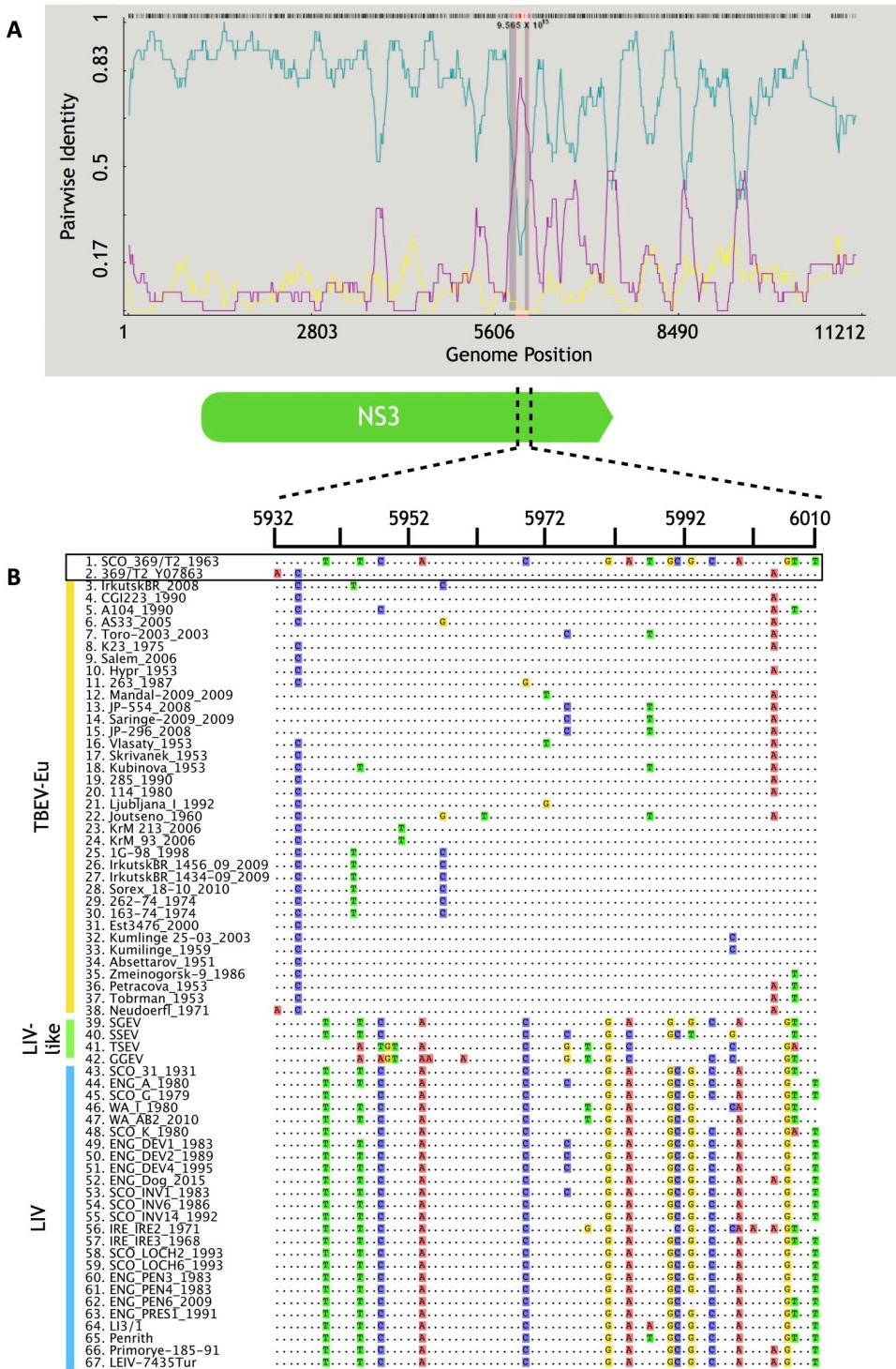

**Fig 2.** A) Recombination detection program (RDP) analysis of the alignment of LIV, LIV-like, TBEV-Eu, TBEV-Sib, and TBEV-FE sequences. A total of 91 genome sequences were included in the analysis. The blue line represented pairwise identity in the GenBank derived LIV 369/T2_Y07863 isolate whilst the purple line represented the pairwise identity of TBEV-Eu strain Neudoerfl. The recombination signal located to the area of intersection, located between nucleotides 5924 and 6129 of the LIV genome. The yellow line represents the LIV isolate IRE_IRE2_1971 which was identified as a potential recombination event however this was not statistically supported and was therefore rejected. B) A simplified alignment of a portion of the suspected recombination region, from nucleotides 5932–6010 for ease of comparison between LIV and TBEV. This is derived from a shortened alignment of 67 genomes which excludes the

TBEV-Sib and TBEV-FE genomes. These TBEV sub-types did not contribute to the observed artefactual recombination signal and are omitted here for clarity. TBEV-Eu, LIV-like and LIV sequences are indicated by vertical bars, placed to the left of the sequence names and coloured yellow, green and blue respectively. The GenBank derived LIV 369/T2_Y07863 isolate, and the re-sequenced SCO_369/T2_1963 genome generated during this study are highlighted. Note that the LIV 369/T2_Y07863 sequence is most similar to the TBEV-Eu sequences, whereas the SCO_369/T2_1963 sequence is most similar to the LIV sequences.

genome is an artefact. Available data for LIV therefore provide no evidence for recombination with other TBFV.

## Genome-wide sequence comparison and selection analysis of LIV

The 26 LIV isolates shared a mean nucleotide identity of ~96% across the entire genome (range: 92.1%–99.9%; Table 1) and a mean amino acid identity of 98.6 (range: 96.9%-99.97%). Average nucleotide identity for individual genes was also ~96%, with NS2A being the most variable (88.4%-91.3% nucleotide identity, 91.3–100% amino acid identity). The amino acid identity for all genes was around ~98%, with NS2B and NS3 most conserved (99.1% amino acid identity). The genetic diversity present within the LIV dataset is similar to that found for TBEV-Eu, with the exception of the 3' UTR which is approximately 96% identical between the LIV isolates, but is highly variable between TBEV-Eu isolates, sharing only around 78% identity (Table 1). Analysis of selection patterns was carried out on the coding region of all 26 LIV genomes (Table 2), using the fixed effects likelihood (FEL) model and the mixed effects model of evolution (MEME). FEL assumes that selection pressures acting on each codon remain constant throughout the phylogeny [64] while MEME allows for positive selection to be episodic and thus to only apply to some branches [51].

Within E, two sites were identified as being subjected to episodic diversifying selection as shown by MEME analysis (Table 2). Of these sites, codon 308 within E encoded a charged aspartic acid residue in all isolates except SCO_INV6_1986, SCO_INV1_1983, ENG_DEV4_1995, ENG_A_1980, and ENG_DEV1_1983, which instead possess a similarly charged glutamic acid. In addition, codon site 100 within NS2A was identified as being under positive selection. Most isolates possessed a glycine residue at this site, however the clade of isolates sampled from southern England (ENG_DEV4_1995, ENG_A_1980, ENG_DEV1_1983, and ENG_DEV2_1989) encodes a serine, as does the Scottish isolate SCO_INV_1983 and the Welsh isolate WA_AB2_2010, whilst the northern English isolate ENG_PEN3_1983 exhibits a cysteine residue.

Codon 96 within NS3 was identified by both MEME and FEL and encoded a threonine residue in the majority of sequences, whilst several isolates throughout the tree, sampled decades apart, from geographically distinct areas, exhibited a methionine, an isoleucine, or a valine. Codon 434 of the NS5 gene was found to encode a histidine in the Welsh isolates, an arginine in IRE_IRE2_1971, a histidine in the oldest sampled Scottish isolate SCO_31_1931 and an alanine in all other isolates (Table 2). As the Welsh isolates, and strains IRE_IRE2_1971 and SCO_31_1931 represent the most ancestral LIV isolates, codon site 434 may therefore have evolved from a histidine to an alanine as LIV spread throughout Britain. SGEV exhibits a histidine at codon 434, further indicating that histidine is the ancestral codon.

Codon 699 within NS5 was also identified as being subjected to positive selection with most isolates encoding an alanine, but with several isolates sampled from different years and locations throughout the tree encoding a valine or threonine. Additionally, FEL identified 482 sites which were under purifying selection (Fig 3). The dN/dS ratio across the entire CDS was found to be 0.0745, indicative of strong purifying selection. Of the individual LIV genes, NS2B

**Table 1. Distances table showing the pairwise genetic and amino acid distances of an alignment of 26 LIV genomes, an alignment of 36 TBEV-Eu genomes, and an alignment combining both datasets.** Mean distances are shown with the range highlighted in brackets. The alignments were generated, and distance tables generated using MUSCLE within the Geneious software package. As some sequences contained gaps at the 5' and 3' genomic termini, only complete genomes were utilised for comparison of the full genome sequence, and the 5' and 3' UTRs. For LIV 21 sequences were utilised for the 5' UTR, 22 for the 3' UTR and 21 for the full genome sequence. For TBEV-Eu 26 sequences were utilised for the 5' UTR, 20 for the 3'UTR, 22 for the full genome nucleotide sequence and 36 genomes for the amino acid sequence.

| | | Identity (%) | | | | Identity (%) | | | | Identity (%) | |
|---|---|---|---|---|---|---|---|---|---|---|---|
| | | Nucleotide | Amino acid | | | Nucleotide | Amino acid | | | Nucleotide | Amino acid |
| LIV | Genome | 96.8 (92.3–99.9) | 98.6 (96.9–99.97) | TBEV-Eu | Genome | 96.5 (94.3–99.9) | 99.1 (98.5–99.97) | LIV/TBEV-Eu | Genome | 91.3 (84.9–99.9) | 96.96 (94.3–99.97) |
| | 5'UTR | 95.4 (83.5–100) | - | | 5'UTR | 97.6 (93.1–100) | - | | 5'UTR | 92.0 (82.6–100) | - |
| | C | 96.9 (91.7–100) | 96.9 (89.3–100) | | C | 98.1 (95.8–100) | 98.6 (96.4–100) | | C | 93.0 (83.9–100) | 94.0 (84.8–100) |
| | prM/M | 96.7 (91.5–100) | 98.9 (97–100) | | prM/M | 96.9 (93.8–100) | 98.5 (94.7–100) | | prM/M | 93.7 (87.5–100) | 96.1 (89.3–100) |
| | E | 96.6 (91.3–100) | 98.6 (96.4–100) | | E | 97.9 (96.6–99.9) | 99.5 (98.2–100) | | E | 92.2 (86.1–100) | 96.6 (92.5–100) |
| | NS1 | 96.6 (91.8–100) | 98.5 (96.3–100) | | NS1 | 98.2 (96.5–100) | 99.1 (97.2–100) | | NS1 | 92.4 (85.6–100) | 97.2 (93.8–100) |
| | NS2A | 96.0 (88.4–91.3) | 97.1 (91.3–100) | | NS2A | 97.0 (94.3–99.9) | 97.6 (94.8–100) | | NS2A | 90.3 (82.3–100) | 94 (87.4–100) |
| | NS2B | 96.6 (90.3–100) | 99.1 (96.2–100) | | NS2B | 98.1 (95.7–100) | 99.4 (96.9–100) | | NS2B | 92.5 (84.2–100) | 99.4 (97–100) |
| | NS3 | 96.6 (91.8–100) | 99.1 (97.7–100) | | NS3 | 97.9 (96.9–100) | 99.5 (98.2–100) | | NS3 | 92.7 (86.6–100) | 97.9 (95.2–100) |
| | NS4A | 96.3 (91.9–100) | 98.9 (96.6–100) | | NS4A | 97.3 (95.1–100) | 99.1 (96–100) | | NS4A | 91.5 (83.7–100) | 97.5 (93.3–100) |
| | NS4B | 96.6 (92.2–100) | 98.7 (96–100) | | NS4B | 97.4 (95.4–100) | 98.7 (96.8–100) | | NS4B | 92.1 (85.3–100) | 96.2 (92.5–100) |
| | NS5 | 96.9 (92.7–100) | 98.9 (97.5–100) | | NS5 | 97.8 (96.7–100) | 99.2 (98.1–100) | | NS5 | 93.0 (87.4–100) | 97.6 (95.4–100) |
| | 3'UTR | 96.2 (91.4–99.8) | - | | 3'UTR | 77.8 (51.4–100) | - | | 3'UTR | 77.5 (51.4–100) | - |

appeared to be under the strongest purifying selection ($\omega = 0.0342$), whilst capsid (C) exhibited the weakest ($\omega = 0.241$).

## Phylogenetic analysis of LIV and closely related TBFV

A Bayesian phylogenetic tree of TBFV genomes placed all LIV isolates into a single monophyletic clade, with SGEV being the closest relative, followed by SSEV and, more distantly TBEV-Eu (Fig 4A). Previous phylogenetic analyses using LIV E gene sequences exhibited only a few nodes with bootstrap values >75% [18], however in our analysis the nodes representing splits between TBFV species all received 100% posterior support as did most splits within LIV, demonstrating the increased phylogenetic certainty which can be achieved using whole genome data (Fig 4B). Within the LIV clade, the Welsh strains WA_I_1980 and WA_AB2_2010 and the Irish strain IRE_IRE2_1971 formed a sister clade to all remaining strains, including all isolates from England and Scotland (86% consensus support).

The tree contained evidence of geographic clustering, even for samples taken decades apart. In addition to the two Welsh strains (1980 vs 2009), this was true for English strains sampled between 1983 and 2009 in the borders region at Penrith (ENG_PEN3_1983, ENG_PEN4_1983, ENG_PEN6_2010) which clustered in a clade that fell close to other strains from the same area, ENG_PRES1_1991 and Penrith. The four isolates sampled from sheep between 1980 and 1995

**Table 2. Alignment of codon sites which were found to be subjected to positive selection.** Codon sites are numbered individually for each gene analysed. FEL denotes sites which have been identified using FEL, MEME denotes sites which have been identified by MEME, and FEL/MEME denotes sites identified by both FEL and MEME analysis. The statistical support for the selected site is given as a p-value, only sites with p < 0.05 are reported. The consensus amino acid is shown for each site and represents the most common amino acid found at that site. Sites which match the consensus are shown as dots, whilst divergent sites display the amino acid which does not match the consensus.

| | Gene | E | E | NS2A | NS3 | NS5 | NS5 | NS5 |
|---|---|---|---|---|---|---|---|---|
| | Model | MEME | MEME | FEL | FEL/MEME | MEME | MEME | FEL/MEME |
| | Codon site | 88 | 308 | 100 | 96 | 434 | 522 | 699 |
| | p-value | 0.040 | 0.051 | 0.043 | 0.021/0.035 | 0.031 | 0.012 | 0.005/0.014 |
| | Consensus | I | D | G | T | A | K | A |
| LIV isolates | IRE_IRE2_1971 | . | . | . | . | R | . | T |
| | WA_AB2_2010 | A | . | . | . | H | . | . |
| | WA_I_1980 | A | . | S | . | H | . | V |
| | SCO_31_1931 | . | . | . | . | H | . | T |
| | Penrith | . | . | . | . | . | . | . |
| | ENG_PRES1_1991 | . | . | . | . | . | . | . |
| | ENG_PEN6_2009 | . | . | . | . | . | . | . |
| | ENG_PEN3_1983 | . | . | C | I | . | . | . |
| | ENG_PEN4_1983 | . | . | . | . | . | . | . |
| | IRE_IRE3_1968 | . | . | . | . | . | . | V |
| | SCO_369/T2_1963 | . | . | . | . | . | . | . |
| | SCO_G_1979 | . | . | . | . | . | . | . |
| | ENG_Dog_2015 | . | . | . | M | . | . | . |
| | SCO_INV14_1992 | . | . | . | . | . | S | . |
| | SCO_INV6_1986 | . | E | . | M | . | . | . |
| | LEIV-7435Tur | . | . | . | M | . | . | T |
| | Primorye-185-91 | . | . | . | M | . | . | T |
| | SCO_LOCH6_1993 | . | . | . | . | . | . | . |
| | SCO_LOCH2_1993 | . | . | . | I | . | . | . |
| | SCO_K_1980 | . | . | . | I | . | . | V |
| | LIV 3/1 | . | . | . | V | . | . | . |
| | SCO_INV1_1983 | . | E | S | M | . | . | . |
| | ENG_DEV4_1995 | . | E | S | I | . | . | . |
| | ENG_A_1980 | . | E | S | I | . | . | . |
| | ENG_DEV1_1983 | . | E | S | I | . | . | . |
| | ENG_DEV2_1989 | . | E | S | I | . | . | . |

from Southern England (ENG_DEV1_1983, ENG_DEV2_1989, ENG_DEV4_1995, ENG_A_1980) also formed a separate clade. There was somewhat less evidence of geographic clustering between the Scottish LIV isolates though most of them fell into a single clade that also contained isolates from other areas. One clade was formed between three isolates sampled from grouse between 1980 and 1993 in the Lochindorb region of Northern Scotland (SCO_LOCH2_1993, SCO_LOCH6_1993, SCO_K_1980). Conversely, three isolates sample near Inverness in the Scottish Highlands did not form a single clade and instead separated into multiple lineages. There were also several cases in which geographically distant isolates grouped together. Specifically, a 1968 Irish isolate (IRE_IRE3_1968), a 2015 isolate from Devon (ENG_dog_2015) and two LIV strains from Far-Eastern Russia (1991) and Turkmenistan (1985), all fell within the larger Scottish clade. This indicates repeated long-distance movement of LIV between Scotland and other countries.

| Gene | Codons spanning genes | Length | dN/dS | Number of sites under selection | | |
|---|---|---|---|---|---|---|
| | | | | FEL | | MEME |
| | | | | Positive | Negative | Positive |
| Capsid | 1-112 | 112 | 0.241 | 0 | 17 | 0 |
| pRM | 113-280 | 168 | 0.064 | 0 | 41 | 0 |
| E | 281-776 | 496 | 0.072 | 2 | 120 | 1 |
| NS1 | 777-1128 | 352 | 0.077 | 1 | 84 | 3 |
| NS2A | 1129-1358 | 230 | 0.161 | 1 | 53 | 2 |
| NS2B | 1359-1489 | 131 | 0.034 | 0 | 36 | 0 |
| NS3 | 1490-2110 | 621 | 0.049 | 2 | 165 | 4 |
| NS4A | 2111-2259 | 149 | 0.082 | 1 | 42 | 2 |
| NS4B | 2260-2511 | 252 | 0.069 | 0 | 60 | 0 |
| NS5 | 2512-3414 | 903 | 0.060 | 2 | 224 | 4 |
| | Total | 3414 | 0.075 | 9 | 842 | 16 |

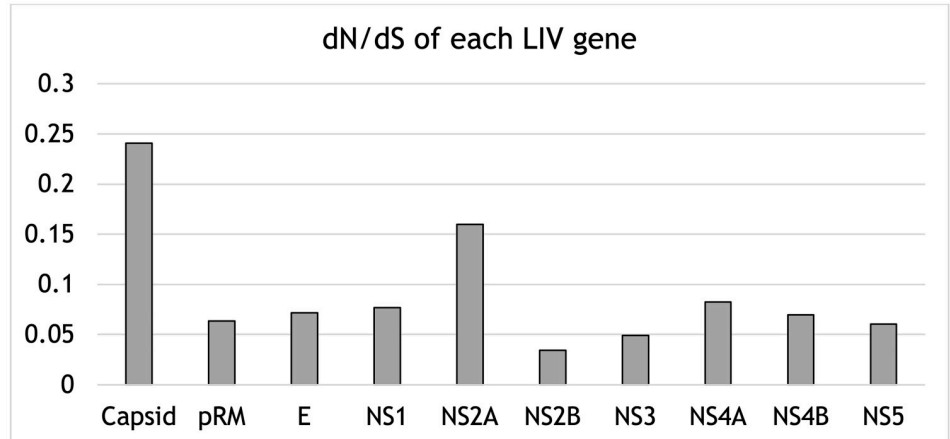

**Fig 3.** A) Gene by gene selection analysis of the 26 LIV genomes included in this study. dN/dS values are shown for each of the LIV genes. A dN/dS value of <1 is indicative of purifying selection. The number of sites under positive and negative selection was calculated using FEL and MEME analysis in the HYPHY software package as part of the DataMonkey web server. B) Graph of dN/dS values for each of the LIV genes.

## Molecular clock analysis reveals a weak temporal signal

Extracting the genetic divergence from the root for all LIV isolates, based on the Bayesian phylogeny, and regressing these values against the sampling dates using TempEst [56] revealed a positive relationship (S1 Fig). This confirmed that the data contain a clock signal, justifying molecular clock analysis in BEAST. The topology of the maximum clade credibility (MCC) tree estimated in BEAST was identical to that of the MrBayes tree (Fig 5). As the presence of strong purifying selection can result in an underestimation of branch lengths, a tree was generated using the adaptive branch-site random effects likelihood (aBS-REL) model [52]. Comparing the total tree length to that of a tree generated under a general time reversible (GTR+G+I) substitution model, revealed no major difference (aBS-REL: 0.38, GTR+G+I: 0.32). This indicates that the branch lengths of the LIV phylogeny were not underestimated as a consequence of purifying selection. To confirm whether a molecular clock rate could reliably be estimated from the LIV data, estimation in BEAST was repeated twenty times with datasets in which sampling dates were randomised. The HPD of the clock rates estimated by all twenty of the

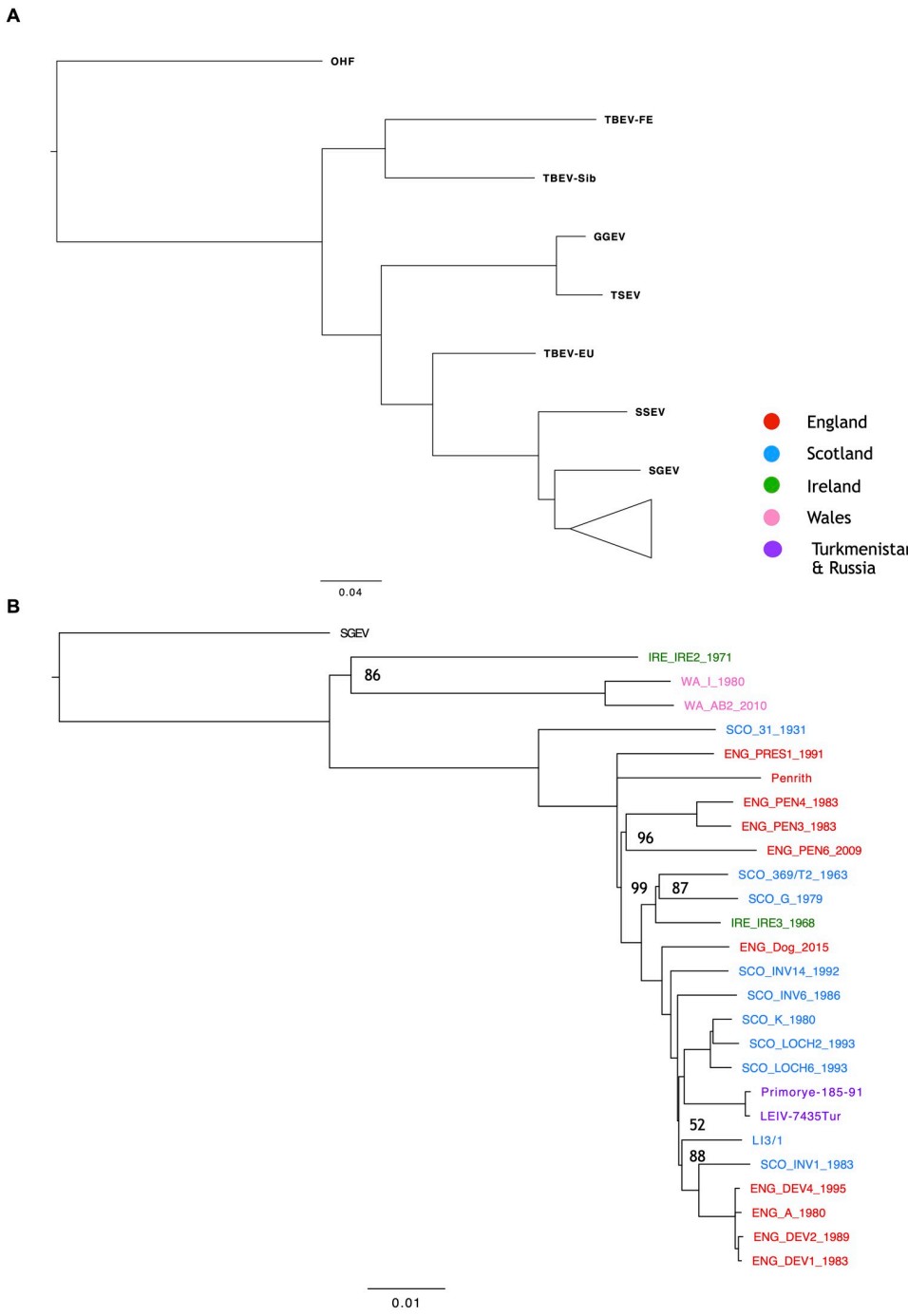

**Fig 4. Bayesian consensus trees generated using MrBayes utilising the 22 LIV genomes generated during this study, four LIV genomes available on Genbank, and eight genomes of closely related BBFs.** A) The Bayesian consensus tree with the LIV clade collapsed, highlighting the relationship of LIV with the other TBFVs included in the analysis. B) The LIV clade alone with SGEV included as an outgroup. The % consensus support value of all nodes was 100, except for those specified. The scale bar represents the number of substitutions per site. The geographic area of isolation of the LIV isolates in B) is denoted by colour. Trees were generated using MrBayes (version 3.2.6, [54]) within the Geneious software suite (version 7.1.9 [40]). The GTR+G+I substitution model with four gamma categories was utilised, as this was found to suit the dataset best using JModelTest [53]. The trees were generated using four heated Markov chain Monte Carlo (MCMC) chains, with a chain length of 1,000,000. Consensus trees were generated using 10% burn-in and a support threshold of 50% and visualised using FigTree (version 1.4.3).

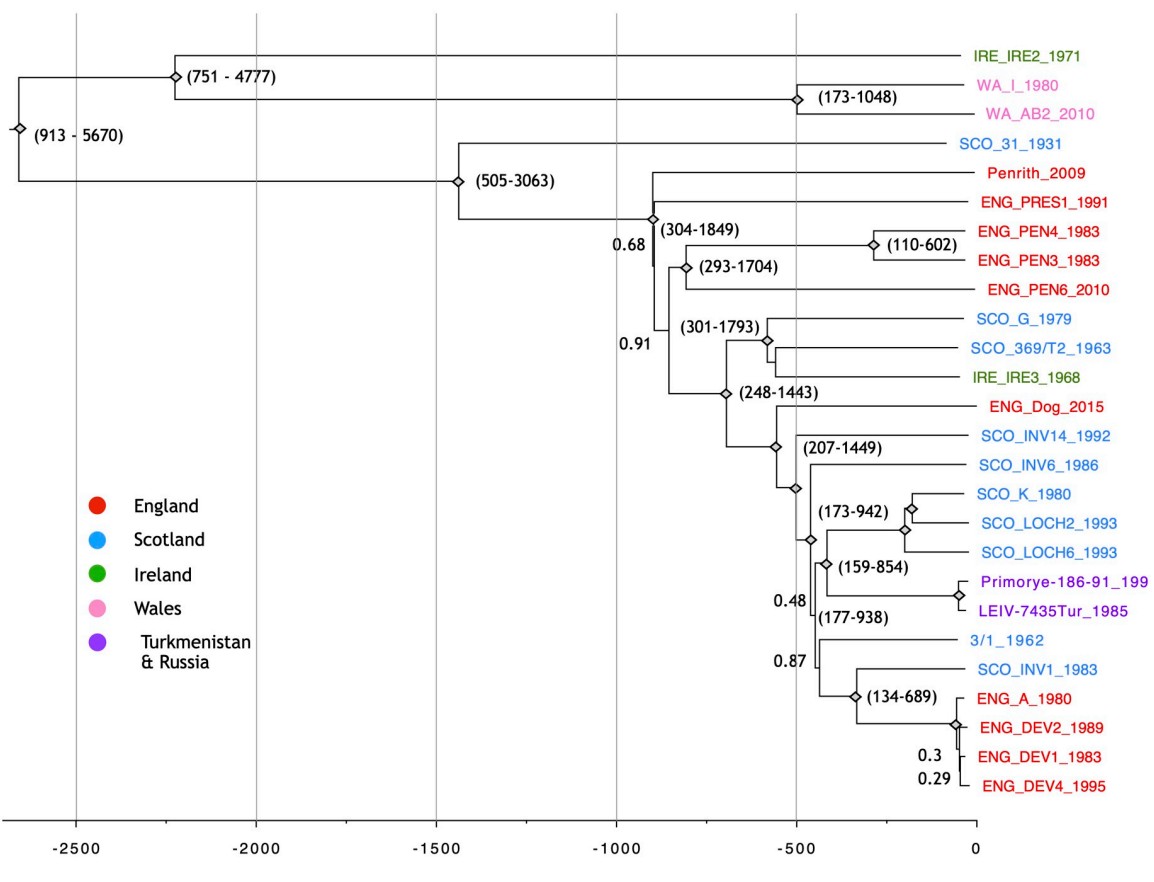

**Fig 5. Time-scaled maximum clade credibility (MCC) tree of 26 LIV genomes collected between 1931 and 2015 estimated in BEAST.** Nodes with posterior probability values of 1.0 are displayed as diamonds whilst values < 1 are shown. The 95% HPD of the node age is shown for selected nodes. However, due to the data set failing a date randomisation test, node age estimates should be considered more uncertain than indicated here. Isolate names are colour coded according to geographic isolation. The dates of isolation of all isolates are included in their tip names.

date randomised BEAST runs overlapped with the mean clock rate estimated from of the original data, indicating that the clock signal in data set was weak and potentially spurious (Fig 6A). The initial BEAST analysis yielded an evolutionary rate of $1.9 \times 10^{-5}$ substitutions/site/year (subst/site/yr; 95% highest posterior density interval (HPD): $5.7 \times 10^{-6}$–$3.9 \times 10^{-5}$ subst/site/yr) and a TMRCA of 3077 years before present (ybp, 95% HPD: 909–5616). Notwithstanding the weak clock signal, this calibration would suggest that most splits in the LIV phylogeny took place hundreds of years ago, with very few nodes having estimate ages within the past century (Fig 5). However, due to potential unreliability of the clock rate, any divergence time inferences should be treated with caution. To test whether the weak molecular clock in the LIV dataset was characteristic for TBFV in general, we estimated a molecular clock and performed date randomisation tests for TBEV-Eu, a close relative of LIV. The clock rate obtained was $3.3 \times 10^{-5}$ subs/site/yr (95% HPD: $2 \times 10^{-5}$–$5 \times 10^{-5}$ substitutions/site/year) which is consistent with previous estimates based on sub genomic datasets [20,32]. Rates estimated for two of the twenty date-randomised runs overlapped with the rate estimated from the observed dataset (Fig 6B).

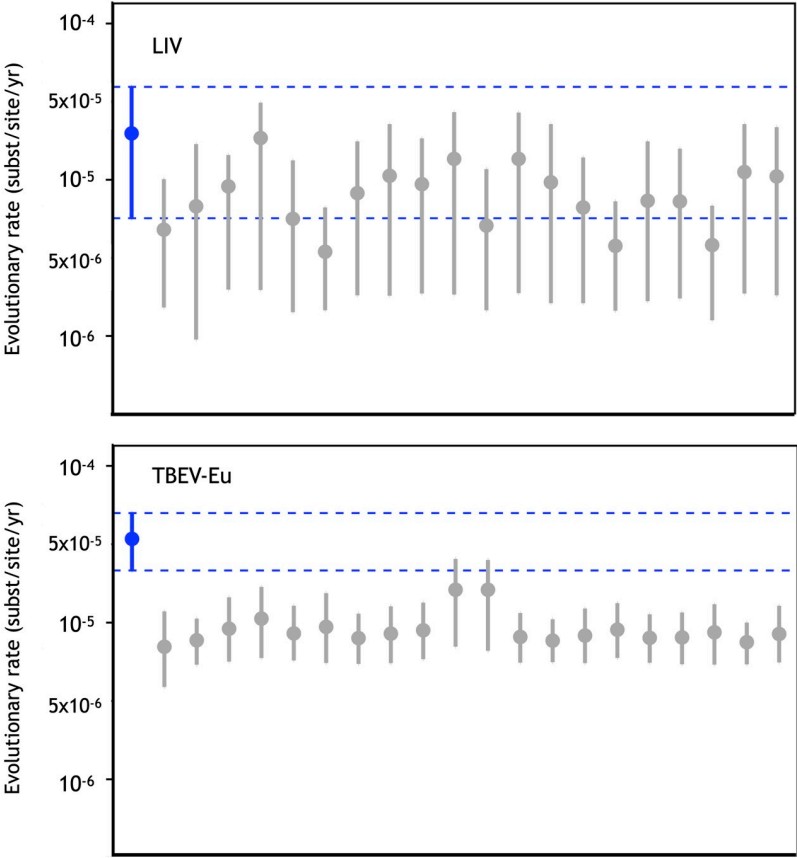

**Fig 6. Plots produced using Tracer which shows the estimated clock rate of the A) LIV and B) TBEV-Eu datasets.**
The rate estimated using the correctly assigned dates are shown in blue while the estimated substitution rate of these datasets using twenty BEAST runs with randomised tip isolation dates are shown in grey. The dotted line highlights runs in which the 95% HPD of the runs overlap.

## Discussion

This study represents the largest evolutionary analysis of LIV to date using a representative set of full genomes sampled across the species' range. Relative to previous analyses, which had been limited to single alignments, this provided new insights into LIV recombination, selection, epidemiology and evolutionary history. No detectable recombination signal was found between LIV and any of the TBFV included. A recombination event between LIV reference strain LIV 369/T2_Y07863 (based on a published genome sequence [1]) and the TBEV-Eu reference strain Neudoerfl had previously been suggested [20,24]. A subsequent study obtained the LIV 369/T2 strain from the European virus archive and, upon re-sequencing the entire genome, found that it differed from the published LIV 369/T2_Y07863 sequence and did not exhibit the previously reported recombination signal [25]. The LIV 369/T2 isolate sequenced in this study was derived from the original virus isolated at the Moredun Research Institute in 1963. We did not find evidence of recombination between LIV and any of the TBFV genomes, thus supporting the findings of Norberg et al. (2013), and their suggestion that the detected recombination signal may be due either to a sequencing artefact or due to an artificial recombination event which took place in a laboratory setting. We therefore conclude that no evidence of past genetic exchange between LIV and TBEV can be detected. Regardless,

recombination is still theoretically possible in areas such as the UK, Norway, and Russia where these viruses now co-circulate due to increasing TBEV range. Such recombination could result in novel TBFV phenotypes of potential public health concern.

While LIV only occasionally infects humans, TBEV is an important human pathogen, therefore a potential LIV/TBEV-Eu recombinant may exhibit altered pathogenicity and present a public health risk. Previous studies have shown that 3'UTR length modulates virulence in some TBEV subtypes [65,66], therefore a LIV-TBEV-Eu recombinant possessing the shorter LIV 3'UTR may possess increased virulence. While it has been demonstrated elsewhere that the deletion of the variable region of TBEV-Eu does not result in increased replication in a cell culture setting [67], this study did not investigate potential differences in pathogenicity using animal models. TBEV-FE exhibits increased virulence compared to TBEV-Eu and TBEV-Sib [5], therefore a recombinant LIV/TBEV-FE virus may also exhibit enhanced pathogenicity. The hypothetical phenotypes of a LIV/TBEV recombinant require confirmation by pathogenesis studies using virus produced by a reverse genetic system. While there is currently no published LIV reverse genetics system, available TBEV reverse genetics approaches could be adapted to investigate the phenotypes of LIV/TBEV recombinants.

Consistent with patterns in other flaviviruses [20,68–72], including TBEV [20,70], the genomic ORF for LIV is subjected to strong purifying selection (dN/dS = 0.0745). Such strong selective constraints are thought to be due to the reliance of the virus on the infection of both vertebrate and arthropod hosts during its life-cycle [71,73]. Only four of the LIV genes (E, NS2A, NS3 and NS5) exhibited any evidence of positive selection. Within E, two sites were identified as being subjected to episodic diversifying selection as shown by MEME analysis (Table 2). Of these sites, codon 308 within E has previously been demonstrated to contribute to monoclonal antibody escape and reduced neurovirulence in mice [74], indicating that substitutions at this site may confer resistance to neutralising antibodies. The functional consequences of the other positively selected sites we have identified are unclear and warrant further study. The selection analysis we have undertaken may inform future vaccine design, as potential vaccine candidates should avoid the inclusion of antigen-determining sites which are subjected to positive selection, as this may influence antibody escape [75,76]. *In silico* analysis of the dataset may identify putative B and T cell epitopes which could be verified by further *in vivo* studies [77–79], however these analyses were beyond the scope of this study.

Genetic diversity present within the LIV dataset was low, with the 26 LIV isolates sharing ~96% mean nucleotide identity and ~98% mean amino acid identity (Table 1). Of the LIV genes, NS2A and C were most divergent, consistent with the weakest purifying selection. Compared with a dataset of TBEV-Eu genomes, LIV appears to exhibit less genetic variability due to the variability present within the TBEV-Eu 3'UTR. The 3'UTR of the combined dataset of LIV and TBEV-Eu genomes shared a mean identity of 77.5% with the most divergent isolate pair exhibiting 51.4% identity. The difference between the 3'UTR of LIV and TBEV is striking and may contribute to the differences in phenotype noted between these viruses. The 3'UTR is responsible for cyclisation during replication but also gives rise to the subgenomic flavivirus RNA (sfRNA) which has been shown to function as a type I interferon antagonist during mosquito-borne flavivirus infection and is involved in pathogenesis [80–82]. In light of the emergence of TBEV in the UK, our dataset could support the development of tools for differential diagnosis between LIV and TBEV. The published LIV and TBEV qPCR methodologies likely cannot distinguish between these closely related viruses [83–85], therefore alternative method employing species specific primers, possibly targeting the 3'UTR, could be developed using our dataset.

Bayesian phylogenetic analysis of the LIV dataset revealed that LIV evolution has been influenced by a mixture of both localised and long-distance transmission events. Within the

tree, distinct sub-clades are present where isolates sampled from similar geographic locations group together. This is particularly evident for the isolates collected in in northern England, northern Scotland, southern England, and Wales. This level of spatial clustering is indicative of localised transmission, where infected animals or ticks are transported over short distances. LIV persistence is also clear in areas such as Lochindorb, Devon, and Penrith, where sub-clades are composed of isolates sampled from the same geographic location decades apart. This implies LIV persistence in these areas and the presence of suitable hosts such as sheep [86,87], mountain hares (*Lepus timidus*) [88,89], red/roe deer (*Cervus elaphus*/ *Capreolus capreolus*) and red grouse [90,91]. Three distinct LIV transmission cycles have been postulated in the UK which rely upon the presence of sheep alone, a mixture of red grouse, mountain hares and deer, and other combinations of these hosts [92,93]. Whilst sheep and grouse support sufficient viremia to infect feeding ticks, mountain hares have been shown to facilitate non-viraemic transmission wherein infected ticks can infect naïve ticks which co-feed on the same animal [89,94]. While deer are not susceptible to LIV and do not support non-viraemic transmission, they are able to carry significant tick burdens and are responsible for amplifying tick populations which, in turn, contributes to LIV persistence [93,95]. One of these proposed transmission cycles must be at work to maintain the virus in areas where LIV genetic lineages remain detectable over multiple decades.

There are also several cases where closely related isolates were sampled from geographically distant areas. One such example is the clustering of the Irish isolate IRE_IRE3_1968 with the western Scottish strains. Whilst this had been previously suggested based on E gene sequence data [18], our phylogenetic placement of IRE_IRE3_1968 based on whole genome data was supported by 100% node support. This provides more confidence in the interpretation that LIV has been potentially (re-)introduced to Ireland from western Scotland, however, it is difficult to draw conclusions as only two Irish LIV sequences are present in the dataset. Additionally, the isolate obtained from a recent canine case in Devon [34], ENG_DOG_2015, appears to be more distantly related to other isolates sampled from the same area and instead clusters between the strains isolated from Western Scotland and SCO_INV1_1983, a strain isolated from Inverness-shire in northern Scotland. Similarly, SCO_INV2_1983 shares a common ancestor with the isolates sampled in Devon and does not cluster with the two other LIV isolates, SCO_INV6_1986, and SCO_INV14_1992, which were also isolated in the Inverness area. This indicates movement of LIV between northern Scotland and southern England.

The most extreme cases of long-distance dispersal in our data are two non-British LIV strains which have been previously isolated in Turkmenistan and the Russian Far East. The cluster containing these two viruses is nested within the UK isolates, indicating that they are direct descendants of the latter, which is puzzling given the large distances involved. While the long-distance movement of LIV within the UK may be explained by sheep trade, with transported animals harbouring infected ticks, the movement of LIV from the UK to Asia and Russia is more difficult to explain. It has been theorised previously that, similar to Powassan virus (POWV), LIV was introduced to Primorsky Krai in the Russian Far East via animal trade following World War I or II [17,96]. While it is not possible to accurately estimate the divergence time of the Russian and Turkmenistan LIV isolates using present data, trade between Vladivostok city, the primary trade hub of Primorsky Krai, and the rest of the world started in the 1860s [96]. Trade with the Russian Far-East was also bolstered by the construction of the trans-Siberian railway which began in 1891 and was completed in 1916 [97]. It can therefore be postulated that if LIV was introduced to Primorsky Krai via trade with the UK, it most likely would have been from the 1860s onwards.

Alternatively, LIV may have been introduced to Turkmenistan and Russia via the transport of ticks by migratory birds. The transport of tick-borne pathogens between countries by birds

has been well documented [98–100], as is the long-distance dispersal of avian influenza viruses by migratory waterfowl [101–103]. However, there are no direct migratory links between central/northeast Eurasia and the UK [104] and in order to transport LIV, birds must reside in environments where LIV is prevalent, such as upland moors, long enough for infected ticks to attach. Moreover, it is implausible that ticks would remain attached to the host over travel distances of several thousand kilometres. Given that LIV has also been detected in Norway and Denmark, it is possible that these countries represent stop-over points where UK strains of LIV have become established locally, and from where they can be distributed further. However, without further investigation these hypotheses are merely speculative.

The hosts which contribute to LIV transmission outside the UK, with the exception of sheep, are currently unknown. As an arbovirus, the range of LIV relies on the presence of its tick vector and susceptible hosts. Unlike TBEV, which is associated with woodland areas where its rodent reservoir hosts are present [105–107], LIV distribution in the UK appears to be predominantly associated with the presence of sheep, red grouse, deer, and mountain hares. As some of these species, or close relatives, are present in Denmark, Norway, and Far Eastern Russia, host communities similar to those in the UK could contribute to LIV transmission in these foci. Interestingly, the Russian isolates were derived from *Ixodes persulcatus* [17], which demonstrates that LIV can be spread by tick species other than its European vector, *I. ricinus*. As these vectors and the main transmission hosts of LIV are distributed across much of Eurasia, it is entirely possible that LIV is present, but remains undetected, in many countries outside of the UK.

Molecular clock analyses indicate that even genomic data from isolates collected over multiple decades might not allow the reliable calibration of a molecular clock for the TBFV. Using date-randomisation, we obtained rate estimates in BEAST that were generally lower but statistically indistinguishable from the observed rate based on the original data. In the meantime, clock rates and divergence time estimates for LIV must be considered potentially unreliable. BEAST analysis produced a molecular clock rate of $1.9 \times 10^{-5}$ substitutions/site/year (95% HPD: $5.7 \times 10^{-6}$–$3.9 \times 10^{-5}$ substitutions/site/year). This clock rate is about an order of magnitude slower than the estimated molecular clock rate of TBEV-Eu [20,31,32] and for LIV based on E gene data only [18] but similar to the rate estimated for Far-Eastern POWV strains [72,96,108]. The estimated TMRCA of the LIV sequences was 3077 years ago (95% HPD: 909–5616), an order of magnitude older than the TMRCA estimated previous based on E gene sequences [18]. Our results suggest that the molecular clock signal in LIV is weak and that larger datasets will be required to achieve more reliable temporal calibration. Future studies should focus on the acquisition of more LIV genomes from within the UK in addition to foci in other countries where LIV has been shown to circulate. Longitudinal sampling of LIV over a number of years from isolated foci, such as some of the Scottish Isles, where it should be possible to sample the same viral lineages through time, would also be useful for the calibration of a molecular clock. This strategy has previously been utilised to investigate the rate of TBEV-Eu evolution [20]. Repeating similar analyses for a dataset of TBEV-Eu genomes revealed that the weak clock signal applied in this case as well, though to a lesser degree. Combined with the increasing recognition that evolutionary rates of viruses are dependent on the time scale of sampling [33], this argues that current estimates of origins and divergence times both within and among TBFV species should be treated with a great degree of caution.

Whilst passage in cell culture could obscure the relationship between genetic distances and sampling dates, this seems unlikely in our case, as the isolates were passaged a maximum of four times; likely too few passages to incur sufficient mutations to confound the clock analysis. Furthermore, strains which exhibit genetic variability dissonant with their year of isolation should be identifiable as outliers in TempEst [56], and no such outliers were apparent.

Likewise, the clinical isolates ENG_Dog_2015, WA_AB2_2010 and ENG_PEN6_2009, which were not passaged in cell culture, did not appear as outliers relative to strains that had been passaged more often. The absence of a strong molecular clock signal in these viruses is therefore more likely to be related to aspects of their biology. Possible factors include the reliance on the tick vector, with potentially long intervals between transmission events, and the alternation between vertebrate host and vector. It is unclear how the rates of viral replication and evolution within ticks, which spent the majority of their life at ambient temperature, compare to those within endotherm mammalian hosts. However, it has previously been reported that TBEV is more prevalent in engorged ticks removed from humans and animals than in wild questing ticks [109–113]. Furthermore, it has been demonstrated that the viral load of a TBEV-Eu strain increased 500-fold after feeding in *in vitro* infected ticks, indicating that blood feeding may affect the viral replication rate within the tick vector [114]. This apparent variation between the replication rate within vertebrate host and vector species may contribute to the weak temporal signal present in the LIV and TBEV-Eu datasets, though it wouldn't explain the apparent difference in their evolutionary rates.

In summary, we have investigated the evolution of LIV using a dataset of 26 LIV genomes, 22 of which were newly sequenced, which were isolated from across the UK in a time period spanning eight decades. Utilising this dataset in addition to 65 genomes from closely related TBFV we did not find evidence of recombination contributing to LIV evolution. While this has been previously thought to be due to the geographic isolation of LIV compared to the other members of the TBFV sub-family [115,116], the identification of LIV foci in Norway, Denmark and Russia represent clear opportunities for recombination to take place. We have confirmed that, like other arboviruses, LIV is predominantly subjected to strong purifying selection however we have identified several sites within the LIV genome which are positively selected for, including one site in the E gene which has previously been implicated in neutralising antibody-escape [74]. Phylogenetic analysis of the LIV genomes clarified the evolutionary relationships of the LIV isolates and indicate that LIV is spread by both localised transmission events within the UK and long-distance dispersals between geographically distant parts of the UK and between the UK and Russia/Turkmenistan. We found that we were unable to accurately estimate the LIV molecular clock, despite utilising a dataset of 22 LIV genomes samples over a period of almost 80 years. This problem also extended to TBEV-Eu and likely also applies to other TBFV (e.g. POWV; [72,108]). The ability to study the evolution of tick-borne viruses is pivotal in our preparedness against these globally emerging pathogens, particularly as evidence for their introduction into new geographic areas continues to be found [27–29]. Our population genomic study of LIV lays an important foundation for further work and highlights the limitations of estimating viral molecular clock rates in TBFV based on current data.

## Supporting information

**S1 Fig. LIV root-to-tip plot of the genetic distance against sampling time of the LIV phylogeny.** Genetic divergence was based on a ML tree generated using the dataset of 26 LIV genomes. The genetic distances from the root to the tips of the ML tree are plotted against the year of isolation. The regression line is plotted according to the best fitting root which minimises the sum of the squared residuals from the regression line. The x-intercept of the regression line represents the TMRCA, while the gradient of the line represents the clock rate. The correlation coefficient ($R^2$) estimates the dispersion of the residuals around the regression line. The plot was generated using TempEst [56].
(TIFF)

**S1 Table. List of virus sequences utilised for phylogenetic analysis.**
(TIFF)

**S2 Table. LIV sequencing statistics.**
(TIFF)

## Acknowledgments

We thank Dr. David Griffiths for providing the CPT-Tert cell line.

## Author Contributions

**Conceptualization:** Colin J. McInnes, Alain Kohl, Roman Biek.

**Data curation:** Jordan J. Clark.

**Formal analysis:** Jordan J. Clark, Roman Biek.

**Funding acquisition:** Colin J. McInnes, Alain Kohl, Roman Biek.

**Investigation:** Jordan J. Clark, Janice Gilray.

**Methodology:** Jordan J. Clark, Roman Biek.

**Resources:** Richard J. Orton, Margaret Baird, Gavin Wilkie, Ana da Silva Filipe, Nicholas Johnson.

**Supervision:** Colin J. McInnes, Alain Kohl, Roman Biek.

**Validation:** Jordan J. Clark, Roman Biek.

**Writing – original draft:** Jordan J. Clark, Roman Biek.

**Writing – review & editing:** Jordan J. Clark, Janice Gilray, Richard J. Orton, Ana da Silva Filipe, Nicholas Johnson, Colin J. McInnes, Alain Kohl, Roman Biek.

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
