## [Decision Letter · Decision Letter 0]

7 Apr 2020

Dear Dr Clark,

Thank you very much for submitting your manuscript "Population genomics of louping ill virus provide new insights into the evolution of tick-borne flaviviruses" for consideration at PLOS Neglected Tropical Diseases. As with all papers reviewed by the journal, your manuscript was reviewed by members of the editorial board and by several independent reviewers. In light of the reviews (below this email), we would like to invite the resubmission of a significantly-revised version that takes into account the reviewers' comments. 

Three experts in the field have reviewed your manuscript. You will notice that there is a broad range of opinions and feedback provided by the reviewers. Please evaluate their assessments carefully when you consider how best to revise your manuscript.

We cannot make any decision about publication until we have seen the revised manuscript and your response to the reviewers' comments. Your revised manuscript is also likely to be sent to reviewers for further evaluation.

Sincerely,

Michael R Holbrook, PhD

Associate Editor

Ann Powers

Deputy Editor

Three experts in the field have reviewed your manuscript. You will notice that there is a broad range of opinions and feedback provided by the reviewers. Please evaluate their assessments carefully when you consider how best to revise your manuscript.

Reviewer's Responses to Questions

**Key Review Criteria Required for Acceptance?**

**Methods**

-Are the objectives of the study clearly articulated with a clear testable hypothesis stated?

-Is the study design appropriate to address the stated objectives?

-Is the population clearly described and appropriate for the hypothesis being tested?

-Is the sample size sufficient to ensure adequate power to address the hypothesis being tested?

-Were correct statistical analysis used to support conclusions?

-Are there concerns about ethical or regulatory requirements being met?

Reviewer #1: The authors have used appropriate methods

Reviewer #2: The methods are in general really well done, one minor caveat, with the selection analysis. For selection analysis you should use IFEL as well. In general, I would treat any that are only found in one method with a great deal of suspicion. Especially as they have very different P-values. Only those found in all three methods would be considered statistically significant.

Reviewer #3: Major comments:

The objectives of this study are clearly stated in the introduction of the manuscript and consist in using a set of 22 new complete genome sequences of Louping ill virus to (re) inspect evolutionary aspects such as recombination, positive selection and evolutionary rates, describe the spatial dynamics of LIV and its phylogenetic relationships with the rest of the Tick-born flavivirus group. 

The study design is appropriate overall but does not go deep enough into the analysis in a number of cases. The positive selection analysis is limited to the detection of statistically significant signal with very little downstream investigation, and the phylogeographic perspective on the phylogeny does not involve phylogeographic inference or any type of historic data that could corroborate the authors observations.

Minor comments:

Sequence assembly:

The authors describe that they have been assembling the LIV genomes by mapping to a reference (Y07863) which they later find to be potentially inaccurate. Have they tried re-mapping to a different reference as a control? Also, when the mapped and reference sequences are divergent enough, the reference can bias mapping and consensus determination. Have the authors tried de novo assembly to control for the the accuracy of the consensus they determined using reference mapping? De novo assembly can be problematic when to few reads are available but for those samples with a sufficient number of reads this could probably be done.

Recombination analysis:

In the recombination screening section, the methods are described but there is no information on the criteria for considering a recombination event as "true". Could you please specify these criteria? For example "a p-value < 0.05 with more than 3 methods was considered as a robust recombination signal". Please also provide the results obtained for the novel sequence (e.g. a graph as a supplementary data).

**Results**

-Does the analysis presented match the analysis plan?

-Are the results clearly and completely presented?

-Are the figures (Tables, Images) of sufficient quality for clarity?

Reviewer #1: Results are relevant and clearly presented

Reviewer #2: I have a few minor issues with the tables, in their presentation otherwise all the results are well presented.

Table 1, identities are not equal throughout the whole data set, please make clear in the legend that these are the ranges and not the same for each virus.

Table 2 – legends do not reflect the table, please correction Me is not used M is.

Reviewer #3: The analysis presented matches the analysis plan. However, the authors do not go deep enough into the analysis in some cases. Investigating further the sites with evidence of positive evolution or digging for historical data corroborating the authors' observations regarding the phylogeography of LIV would be most valuable for the study. On the spatial dynamics results, several of the observations made by the authors rely on a small number of sequences and are not necessary in line with the data present in the tree. Please, re-write that section more cautiously considering that you do not have a very important sampling over time and space.

Major comments:

Selection analysis:

The selection analysis is limited to identifying statistically significant signal using to different methods with no further investigation. The results are fairly descriptive with no insight on possible functional evolution. For sites with statistically significant signal of positive selection it would be very valuable to go deeper into the analysis. A few examples of the questions that would be worth asking:

Are mutations a the site occurring several times in the phylogeny? 

Is it only in present in recent sequences?

Is it occurring in a conserved area of the genome?

Is there literature on this specific mutation/site/region of the genome?

When you have a deep enough coverage do you have population mixture at this site?

For example, the mutation in NS3, occurs in different parts of the tree, at different 

points in time, occurs in a region with low dN/dS...

Phylogeny:

Overall, I do not agree with a number of the conclusions of the authors on the phylogenetics/spatial dynamics section:

l356-357: "The tree contained repeated evidence of geographic clustering, even for samples taken decades apart." 

There are several exceptions to this statement, the sequences from Ireland do not cluster together,

the sequences from southern England do not cluster all together (ENG_Dog_2015 is not clustering with the rest of the sequences A, DEV1/2/4), the sequences from northern Scotland do not all cluster together either (INV1 is separate form INV6 and INV14).

l359-362: "SCO_LIV_31 sampled in 1931 from the Scottish borders, which grouped with English strains sampled more recently in the borders region (Penrith, ENG_PRES1_991, ENG_PEN3_1983, ENG_PEN4_1983,ENG_PEN6_2010)."

SCO_LIV_31 is still quite distant from the English strains mentioned above.

l362-363: "All other Scottish strains (sampled between 1962 and 1993) formed a separate 

clade"

That clade also includes sequences from the south of England (ENG_Dog_2015) and from Ireland (IRE3), which make up to a third of the clade.

One important statement that can be made from this tree (and that is made by the authors) is that the sequences from eastern Russia and Turkmenistan probably have an origin in the UK/Ireland. This is a very interesting point to discuss in terms of historical data that may support or contradict this observation.

Molecular clock analysis:

As it is, the order of the paragraph is surprising to me because the authors use theevolutionary rate they infer to evaluate tMRCAs for the whole species and ancestral nodes before testing (aBSREL model and randomisation) the robustness of their results. 

Mentioning the tMRCA (if at all) should be done way more cautiously knowing that the evolutionary rate that was used for inference is not reliable.

It is not clear to me why, since you show that your estimate may not be reliable since there is an overlap with the estimates obtained using sequences with randomized dates, you repeat the analysis with the CDS without using a date-randomisation control.

Minor comments:

l352-353 "increased phylogenetic certainty resulting from whole genome data"

Please illustrate your point (e.g. with a comparison) stating for which species and in which studies the split was uncertain.

**Conclusions**

-Are the conclusions supported by the data presented?

-Are the limitations of analysis clearly described?

-Do the authors discuss how these data can be helpful to advance our understanding of the topic under study?

-Is public health relevance addressed?

Reviewer #1: Conclusion are within bounds in relation to the results

Reviewer #2: The discussion is robust, it could have been a little more expansive on the 3'UTR which seems to be the most intriguing difference between LIV and TBEV, but that is my opinion and does not need to be changed.

Reviewer #3: Some of the conclusion are in line with the data, others are not. The predominance of localised transmission in the dispersal dynamics is not clear based on the trees that are presented by the authors and rather suggest a mixture between localised and long-distance transmission. The authors also discuss the data based on molecular clock estimates before mentioning that their estimates are not reliable, which is misleading for the reader.

One of the limitations of this study is the small number of sequences with regard to the analysis. The LIV dataset is currently too small, with no longitudinal sampling to accurately describe the local dynamics of the spread of LIV.

Specific comments:

l457-458: "Such recombination could result in novel TBFV phenotypes of potential public health concern."

Please develop here. Is there experimental or field evidence of specific lineages/species being more virulent, transmissible (or other phenotype relevant to public health) than others?

"Our findings could support the development, if necessary, of tools for differential diagnosis and vaccine design." 

Same here, please develop on how and why. For example, you could discuss the utility of the information on diversity and purifying selection analysis for vaccine and drug development.

l472-473: "Codon site 100 within NS2A was identified as being under positive selection as was codon 96 within NS3."

l480-481:"Sites 522 and 699 within NS5 were also identified as being subjected to positive selection."

Here you're simply stating your results, please develop on this statements or remove.

l479-480: "SGEV exhibits a histidine at codon 434, further indicating that histidine is the ancestral codon."

Please develop on the possible functional consequences of histidine being the ancestral codon, otherwise the relevance of this information is not clear.

l496-498:"Bayesian phylogenetic analysis of the LIV dataset revealed that the Irish and Welsh isolates tended to cluster separately from the English and Scottish isolates, with the exception of IRE_IRE3_1968."

Since there are only two Irish sequences, only the Welsh isolates cluster separately then.

l496-519: I don't agree with most of the paragraph, there is possibly a degree of local transmission but there are also movements between Ireland and the UK, North and South and given the current sampling and data it is not possible conclude that one type of transmission is predominant over the other.

On the introduction from western Scotland to Ireland please be more cautious since there are only three sequences sampled several years apart.

"from scotland to england" 

Here, there is no basis for assuming any directionality in the movements between England and Scotland with certainty.

line 520-529:

Here it would be very insightful to have historical data on exchanges between the UK and Eastern Russia/Turkmenistan to further inform that part of the tree and possibly clarify the timing of the introduction.

l530-535: Here I do not agree with the structure of the paragraph. The authors should start by making clear that they identified that the estimates are not reliable and then discuss the results in order to identify how the estimates could be improved or corroborated.

l574-576:"This apparent variation between the replication rate within vertebrate host and vector species may contribute to the weak temporal signal present in the LIV and TBEV-Eu datasets" 

Yes, and please discuss the other limitations that could explain the lack of temporal signal here.

l581: "recombination has not contributed to LIV evolution"

You could be more cautious here because this is true, given the currently sampled viral diversity.

582: "geographic isolation of LIV" I'm not so sure about that.

590: "predominantly by localised transmission events with few long-distance

590 dispersals." Again, I do not agree with this conclusion.

**Editorial and Data Presentation Modifications?**

Reviewer #1: (No Response)

Reviewer #2: Line 303, need to amend as this is the wrong tense

I would also not focus as much on the recombination, I think allowing this idea to fade away would be a really good thing, and so I would not focus on it other than to mention that you have eliminated the possibility due to previous sequencing errors.

Reviewer #3: In the virus isolates section, I do not seem to find any description for isolation/sampling of strains ENG_PEN6_2009 and WA_AB2_2010.

Please introduce abbreviations (e.g. TBEV-Eu, TBEV-Sib, TBEV-FE)

l288-291: "The genetic diversity present within the LIV dataset is similar to that found for TBEV-Eu, with the exception of the 3’ UTR which is approximately 96% identical between the LIV isolates, but is highly variable between TBEV-Eu isolates (Table 1)."

Please provide the %id for the 3'UTR region in the TBEV-Eu group to illustrate your point and ease reading.

Some dates are not matching between Fig 1, and the trees. Is WA_AB2 from 2009 (tree-Fig5) or 2010 (table-Fig1)?

Please provide access to raw sequencing data.

**Summary and General Comments**

Reviewer #1: Review of manuscript PNTD-D-20-00207: "Population genomics of louping ill virus provide new insights into the evolution of tick-borne flaviviruses" by Jordan J. Clark et al.

The authors have studied the spread and evolution of a virus which, before this study, has not received a lot of attention despite its relevance to animal health. They also present data on molecular dating in relation to other tick-borne flaviviruses. The results are likely of interest to a broader community of readers. Below are my comments to their manuscript. I hope they will help to clarify certain parts and improve their manuscript.

In general I only have minor comments. I think the manuscript is well-written and clear in most aspects. It is difficult to have everything in one manuscript, but what I find to be missing is a discussion around the ecology, host-association and geographical distribution of LIV.

For example, given that it seems to be fairly geographically restricted (although it is found at very geographically different locations), it would be very interesting to hear the authors ideas as for what is needed for LIV to become established in a particular area? Why is it relatively prevalent in the UK but (seemingly) not elsewhere? What habitat and ecological circumstances are necessary for LIV to become established in one area? Is it simply likely to be undersamples elsewhere? 

Although the focus is clearly the evolutionary history of LIV/TBFV, I think a paragraph or so regarding the tick-virus-reservoir host-ecology relationship would complement the manuscript quite well. 

Minor comments:

-r71: "...resulting in febrile illness and often fatal encephalitis (4)..."

"Often" is relative of course. But I would say that it makes more sense to mention e.g. incidence / case fatality rate.

-r141: "..).RNA.." Add space.

-r153-160: I would probably use e.g. "sequence libraries" or similar instead of "samples" to distinguish between RNA samples and libraries.

-r214: "suite(37,52)" add space

-r283 and more: Check usage of range. "92.1% – 99.9%" vs. "92.1%–99.9%"

Reviewer #2: This paper is a discussion of the evolution of LIV and highlights the similarities and differences from TBEV.

The paper is coherent and has some excellent points. I have a few minor comments that I feel need to be addressed but the paper has been well designed and the analyses are in general robust.

Reviewer #3: In this study the authors sequence complete genomes from a set of 22 Louping ill virus isolates and use them to (re)inspect evolutionary aspects such as recombination, positive selection and evolutionary rates at the level of the species. 

Using RDP, they identify no evidence of recombination within the LIV species and find additional support for the absence of recombination between the isolate LIV 369/T2_Y07863 and sequences from the TBEV Eu group. Using two different maximum likelihood approaches they find evidence of positive selection within the LIV species at seven sites and show broad purifying selection with low dN/dS

ratios observed in most regions of the genome. The authors use a combination of ML and 

Bayesian tree-building methods to infer the phylogenetic relationships between LIV isolates and between LIV species and the rest of the TBFV group. They use these trees, in combination with space and time information attached to the sequences to identify both localised and long-distance transmission dynamics of LIV in the UK and Ireland. Finally, using Bayesian inference they infer the evolutionary rate within the species and identify that their estimate is not reliable, using a control dataset with randomised dates.

This study is robust in terms of methodology but has several limitations in terms of analysis. The selection analysis dos not go deep enough so that this part of the results is only descriptive with no insight into possible functional evolution within LIV. The conclusions of the authors on the spatial spread of the virus do not reflect what is shown in the trees, identifying a predominantly local transmission while the trees rather point at a mixture of local and long-distance transmission with no dynamic being clearly predominant. Also, given the sampling, it is difficult to confirm local transmission, longitudinal sampling would be ideal here. Finally, the authors should make clear in their results and in the discussion section that the rate estimates they discuss are not reliable. In the current form, they start by discussing the results before mentioning that the evolutionary rates are not robust.

PLOS authors have the option to publish the peer review history of their article (what does this mean?). If published, this will include your full peer review and any attached files.

Reviewer #1: No

Reviewer #2: No

Reviewer #3: No
---

## [Decision Letter · Decision Letter 1]

17 Jul 2020

Dear Dr Clark,

Thank you very much for submitting your manuscript "Population genomics of louping ill virus provide new insights into the evolution of tick-borne flaviviruses" for consideration at PLOS Neglected Tropical Diseases. As with all papers reviewed by the journal, your manuscript was reviewed by members of the editorial board and by several independent reviewers. The reviewers appreciated the attention to an important topic. Based on the reviews, we are likely to accept this manuscript for publication, providing that you modify the manuscript according to the review recommendations. 

Our reviewers have re-evaluated your submission and found that the concerns raised after the first review were generally adequately addressed. One reviewer did have some minor comments that they felt would improve the presentation of your study. Once these comments are dealt with, we should be in good shape to move forward with your submission.

Sincerely,

Michael R Holbrook, PhD

Associate Editor

Ann Powers

Deputy Editor

Our reviewers have re-evaluated your submission and found that the concerns raised after the first review were generally adequately addressed. One reviewer did have some minor comments that they felt would improve the presentation of your study. Once these comments are dealt with, we should be in good shape to move forward with your submission.

Reviewer's Responses to Questions

**Key Review Criteria Required for Acceptance?**

**Methods**

-Are the objectives of the study clearly articulated with a clear testable hypothesis stated?

-Is the study design appropriate to address the stated objectives?

-Is the population clearly described and appropriate for the hypothesis being tested?

-Is the sample size sufficient to ensure adequate power to address the hypothesis being tested?

-Were correct statistical analysis used to support conclusions?

-Are there concerns about ethical or regulatory requirements being met?

Reviewer #1: The methods are well-executed, -described and -motivated.

Reviewer #2: All my comments were adequately addressed.

Reviewer #3: The authors answered all the comments regarding the method section and corrected the manuscript wherever it was necessary.

Minor comments:

(1) Whenever de novo assembly is possible I would recommend always controlling the consensus sequence by comparing what is obtained with de novo assembly versus alignment to a reference.

(2) In Figure 2.A, please specify in the legend which sequence corresponds to the yellow line.

**Results**

-Does the analysis presented match the analysis plan?

-Are the results clearly and completely presented?

-Are the figures (Tables, Images) of sufficient quality for clarity?

Reviewer #1: The results and related figures/tables are clearly presented and match the aim of the manuscript.

Reviewer #2: All my comments were adequately addressed.

Reviewer #3: The authors answered all the comments regarding the results section and in most cases, corrected the manuscript wherever it was necessary. I still have some minor comments on that section:

line 350: "MEME denotes sites which have been identified by MEME"

Please correct the legend so it correctly describes what is in the table. In my understanding it should be MEME rather than ME.

On the Selection analysis:

The authors provide a better description of the mutations identified in the selection analysis. I would recommend moving some of the information to the results section (e.g. which strains carry which amino acid at a given position) and in the discussion, have a documented and well-argued paragraph on why the readers should care (or not) about some of the mutations. For now that part of the discussion looks a little too much like a catalogue.

On the Phylogenetic analysis:

The authors have improved their conclusions on the spatial dynamics of LIV but some inaccuracies remain:

line 573-575: "This is particularly evident for the isolates collected in northern England, northern Scotland, Wales and Southern England."

As mentioned in my previous comment, not all sequences from Northern Scotland cluster together (INV1 is separate form INV6 and INV14). The authors should specify this point in the narrative, as they do for the isolate ENG_DOG_2015.

General comment on paragraph 571-593:

While discussing mechanisms of localised transmission, rather than just considering small distance movements, the authors should also discuss possible mechanisms for persistence in a population over the years because this would also be an important part of the phenomenon.

lines 591-608: On strains from Eastern Russia and Turkmenistan:

I completely agree with the authors, if there is no or very little historical data, the authors should not speculate too much. However they can cautiously lay out several possible scenarios specifying that they are only hypotheses.

line 594: Recent is very relative, I recommend avoiding using such qualifier, or specifying in which context it can be considered "recent".

lines 606-608: The last sentence is a little too speculative. I recommend making a general statement stating that based on the (scarce) historical data available, introduction of LIV into these regions might have occurred via trade routes. The authors should avoid specifying a point of entry, because there is no proof for that. Important point, the tree suggests that the lineages in Eastern Russia and Turkmenistan stemmed from the UK region, pinning down the geographical origin of the introduction is not possible given the sampling and the approach used here.

On the minor comment:

line 365-370: The amended version is better but I don't think this is about resolution. For this analysis, genomic sequences were more informative than single gene sequences and yielded more robust results.

Table 2 legend line 350: "ME denotes sites which have been identified by MEME"

Based on what is shown in the table, it should be MEME

**Conclusions**

-Are the conclusions supported by the data presented?

-Are the limitations of analysis clearly described?

-Do the authors discuss how these data can be helpful to advance our understanding of the topic under study?

-Is public health relevance addressed?

Reviewer #1: Conclusions are within bounds of what can be inferred from the data and addressed properly.

Reviewer #2: All my comments were adequately addressed

Reviewer #3: The authors answered all the comments regarding the discussion section and in most cases, corrected the manuscript wherever it was necessary. I still have some minor comments on that section:

On paragraph 484-499:

When writing that there are no reverse genetics systems available for LIV the authors could however mention that reverse genetics systems developped for TBEV could most likely be harnessed for studying the phenotype of LIV/TBEV recombinants.

lines 514-516: "the clade of isolates sampled from ... encode a serine" 

In my understanding it should be "encodes".

line 581 in original manuscript: "recombination has not contributed to LIV evolution" and lines 685-687 in the amended version.

The amended version is not clear enough on that point. I suggest replacing "we have confirmed" with a more cautious formulation "we did not find evidence of recombination contributing to LIV evolution"

On the comment of the authors on the "geographic isolation of LIV". 

I get the point however, there is evidence that LIV is present outside the UK and might have been for some time, as indicated by the identification of LIV in samples from Eastern Russia and Turkmenistan.

**Editorial and Data Presentation Modifications?**

Reviewer #1: (No Response)

Reviewer #2: No changes required.

Reviewer #3: The authors corrected the manuscript wherever it was necessary. On last edit I recommend is increasing the resolution of the figures.

**Summary and General Comments**

Reviewer #1: The authors have addressed all my previous comments. I have nothing further to add.

Reviewer #2: No changes required

Reviewer #3: The authors answered all the comments from reviewers #1, #2, and #3 and made most of the corrections that were recommended. The manuscript as greatly improved so that only a few minor modifications are now necessary for it to be ready for publication.

PLOS authors have the option to publish the peer review history of their article (what does this mean?). If published, this will include your full peer review and any attached files.

Reviewer #1: No

Reviewer #2: No

Reviewer #3: No
---

## [Editor Report · Decision Letter 2]

7 Aug 2020

Dear Dr Clark,

We are pleased to inform you that your manuscript 'Population genomics of louping ill virus provide new insights into the evolution of tick-borne flaviviruses' has been provisionally accepted for publication in PLOS Neglected Tropical Diseases.

Best regards,

Michael R Holbrook, PhD

Associate Editor

Ann Powers

Deputy Editor

---

## [Editor Report · Acceptance letter]

8 Sep 2020

Dear Dr Clark,

We are delighted to inform you that your manuscript, "Population genomics of louping ill virus provide new insights into the evolution of tick-borne flaviviruses," has been formally accepted for publication in PLOS Neglected Tropical Diseases.

Best regards,

Shaden Kamhawi

co-Editor-in-Chief

Paul Brindley

co-Editor-in-Chief
